# Croargoids A–G, Eudesmane Sesquiterpenes from the Bark of *Croton argyratus*

**DOI:** 10.3390/molecules27196397

**Published:** 2022-09-27

**Authors:** Min Wu, Kai-Long Ji, Peng Sun, Jian-Mei Lu, Jia-Rui Yue, Dong-Hua Cao, Chun-Fen Xiao, You-Kai Xu

**Affiliations:** 1Key Laboratory of Tropical Plant Resources and Sustainable Use, Xishuangbanna Tropical Botanical Garden, Chinese Academy of Sciences, Xishuangbanna 666303, China; 2University of Chinese Academy of Sciences, Beijing 100049, China; 3School of Pharmaceutical Science and Yunnan Key Laboratory of Pharmacology for Natural Products, Kunming Medical University, Kunming 650500, China; 4The Affiliated Changsha Central Hospital, Hengyang Medical School, University of South China, Changsha 410004, China

**Keywords:** *Croton argyratus*, Euphorbiaceae, eudesmane sesquiterpene, NO inhibition

## Abstract

Seven new sesquiterpenes, named croargoid A–G (**1**–**7**), were isolated from the bark of *Croton argyratus*. Compounds **1**–**4** were the first examples of eudesmane sesquiterpene lactones containing C_5_-OH group. Compound **7** was a highly degraded eudesmane sesquiterpene possessing a rare eleven-carbon skeleton. Their structures with stereochemistry were mainly elucidated by NMR analyses in combination with MS and ECD data. Cytotoxicities and NO inhibitions of all isolates were evaluated and only compound **5** showed moderate NO inhibitory activity.

## 1. Introduction

*Croton*, a genus of Euphorbiaceae family, possesses more than 1300 species around the world and are widely distributed in tropical and subtropical regions, of which most are trees or shrubs [1]. Some *Croton* species have a long history of use in traditional medicine in Asia, Africa, and South America, such as in the treatment of cancer, constipation, diabetes, digestive problems, dysentery, fever, high blood pressure, inflammation, intestinal parasites, malaria, and weight loss [2,3]. Previous phytochemical investigations of *Croton* revealed that the major constituents were diterpenoids [4,5,6,7], sesquiterpenes [8], triterpenes [9], and glycosides [10] exhibiting cytotoxic [11], anti-inflammatory [12], and antifungal activities [9]. *C. argyratus* is an important ethnic medicine, mainly distributed in Malaysia, Indonesia, Philippines, Vietnam, Thailand, and other Southeast Asian countries [13]. To search bioactive metabolites with unique structures from medicinal plants, the chemicals of *C. argyratus* was investigated. As a result, seven new eudesmane sesquiterpenes (Figure 1), named croargoids A–G (**1**–**7**), were isolated from the 95% EtOH extract of *C. argyratus* barks. Compounds **1–4** were the first examples of eudesmane sesquiterpene lactones containing C_5_-OH group, which were also the representative eudesmane sesquiterpenes found in the Croton genus. All the isolates were evaluated for their cytotoxicities and NO inhibitory effects. Among them, compound **5** exhibited moderate NO inhibitory effect.

## 2. Results and Discussion

### 2.1. Structure Identification of New Compounds

Croargoid A (**1**), a white amorphous powder, has a sodium adduct ion at *m/z* 273.1463 (calcd. for C_15_H_22_O_3_Na^+^, 273.1461) in the HR-ESIMS spectrum, indicating a molecular formula of C_15_H_22_O_3_ with five degrees of unsaturation (DOUs). Its IR spectrum showed the presence of hydroxyl (3514 cm^−1^) and carbonyl (1735 cm^−1^) groups. Analysis of the ^1^H NMR data (Table 1) of compound **1** indicated the signals for three methyl groups (*δ*_H_ 1.81, *δ*_H_ 1.16, and *δ*_H_ 0.91). Further inspection of the ^13^C and DEPT NMR spectra (Table 2) exhibited the existence of 15 carbon resonances, including four quaternary carbons (which belong to one ester carbonyl, two olefinic, and one oxygenated), two methane (one oxygenated), five methylenes, and three methyls. Taken together, these functional groups accounted for 2 out of 5 DOUs and the remaining ones suggested that compound **1** was tricyclic.

The comparisons of 1D NMR data between compound **1** and herticin A [14] revealed that they were structural analogs and the major differences were the different appendances at C-5 and C-10 positions. In detail, herticin A was a hydroxyl attached to the C-10 and a methyl attached to the C-5, while there was a hydroxyl and a methyl group attached to the C-5 and C-10 position, respectively, in **1**, whereas they were opposite in herticin A. This conclusion was supported by the HMBC correlations of OH-5/C-5, C-6, and C-10; H_3_-14/C-1, C-9, and C-10 (Figure 2).

The relative configuration of **1** was established by the NOESY spectrum. The cross peaks of H-8/H_3_-14, H_3_-14/H-6*β*, H-6*β*/H-4, and H_3_-14/H-4 (Figure 3) demonstrated that these protons were oriented on the same sides and arbitrarily assigned as *β*-orientation, while the cross peaks of H-6*α*/H_3_-15 established that H_3_-15 was *α*-orientation. Finally, the absolute configuration of **1** was determined by comparing its experimental ECD spectrum with the theoretical data (Figure 4). In the 210–270 nm regions, both the experimental ECD spectrum and the calculated one for **1** showed the same positive cotton effect, which determined the absolute configuration of **1** as 4*R*, 5*R*, 8*S,* and 10*R*.

Croargoid B (**2**) was obtained as a white amorphous powder, which possessed a molecular formula of C_15_H_22_O_3_ on the basis of the HRESIMS peak at *m/z* 295.1553 [M+COOH]^-^ (calcd. 295.1551). Its ^1^H and ^13^C NMR data (Table 1 and Table 2) were very similar to those of **1**, with the only difference being at C-8, which suggested that compound **2** was likely the C-8 epimer of **1**. This was verified by the NOESY correlations of H-8/H-9*α* which can be observed and established with H-8 as *α*-direction (Figure 3). In addition, the absolute configuration of **2** (4*R*, 5*R*, 8*S,* and 10*R*) was assigned through ECD calculation (Figure 4).

The molecular formula of croargoid C (**3**) was determined C_15_H_22_O_4_ via the ion peak at *m/z* 265.1446 [M-H]^-^ (calcd. 265.1445) in the HRESIMS spectrum, exhibiting 16 mass units more than that of **1**. The UV spectrum of compound **3** (*λ*_max_ 222 nm) resembled those of **1** (*λ*_max_ 223 nm), together with the similar ^1^H and ^13^C NMR data between **3** and **1** (Table 1 and Table 2) implied that they were structures analogues [15]. The only difference was that compound **3** has one additional hydroxyl group at C-8 (*δ*_C_ 104.1). This was further confirmed by the HMBC correlations of H_2_-6/C-8 and H_2_-9/C-8 but lacks the signals of H-8/C-11, C-12 (Figure 2). Additionally, compound **3** possesses the stereochemistry of 4*R*, 5*R*, 8*S*, 10*R* by using ECD calculation (Figure 4), which was identical to that of **1**.

Croargoid D (**4**) was obtained as a white amorphous powder. Its molecular formula C_15_H_20_O_3_ was established by the HRESIMS spectrum, having six DOUs and 2 mass units less than **1**. The NMR data of 4 (Table 1 and Table 2) were similar to those of **1** with the presence of an additional double bond between C-8 and C-9. The observed HMBC cross peaks of H-9 (*δ*_H_ 5.38)/C-7, C-5, and H_3_-14/C-1, C-9, and C-5 further supported the above assignment (Figure 2). The absolute configuration of **4** was finally assigned by comparison of the experimental ECD spectrum, the calculated spectra showed the same trend as the experimental one (Figure 4), indicating that the 4*R*, 5*R*, and 10*R* for compound **4**.

Croargoid E (**5**) was obtained as a white amorphous powder. Its molecular formula C_15_H_24_O_2_ with four DOUs was established by the HRESIMS ion peak at *m/z* 259.1668 ([M+Na]^+^, calcd. 259.1669). In the IR spectrum, the absorption bands at 3434 cm^−1^ suggested the presence of a hydroxyl group. The 1D NMR data revealed that compound **5** has one keto (*δ*_C_ 203.1), four quaternary carbons (two olefinic), one methine, five methylenes, and four methyls. Those functionalities account for 2 out of 4, suggesting compound **5** was a bicyclic compound.

Further analysis of the 2D NMR spectra could establish the structure of **5** (Figure 3). A spin-spin coupling system of H_2_-1/H_2_-2/H_2_-3/H-4/H_3_-15 was observed in the ^1^H-^1^H COSY spectrum, while the connection with other atoms was established by HMBC spectrum. The HMBC correlated system (H-1/C-14; H-6/C-8, C-10, and C-11; H-9/C-5, C-7, and C-14; H_3_-12/C-7; H_3_-13/C-7, and C-12; H_3_-14/C-5 and H_3_-15/C-5) construct the planar structure of **5** as a bicyclic eudesmane sesquiterpene with the loss of ring C in compound **1** and the presence of an exocylic *α*, *β* -unsaturated keto at the C-11–C-7–C-8 position. In the ROESY spectrum, the cross peaks of H_3_-14/H-4 and H-6*α*/H_3_-15 indicated that these protons were co-facial and assigned *β*-orientation, while the cross peak of H-6*α*/H_3_-14 determined *α*-orientation for H_3_-15. In addition, the absolute configuration of **5** was finally determined as 4*S*, 5*R*, and 10*S* by the comparisons between experimental and calculated ECD data (Figure 4).

Croargoid F (**6**) was obtained as a white amorphous powder. Its molecular formula C_15_H_24_O_2_ was established by the HRESIMS ion at *m/z* 259.1668 ([M+Na]^+^, calcd. 259.1669). Analysis of the 1D NMR spectra of **6** and **5** (Table 1 and Table 2) indicated that they shared an identical carbon skeleton. The main difference was the presence of an additional hydroxyl at C-4 in compound **6**. This conclusion was supported by the downfield carbon signals of C-4 from *δ*_C_ 35.0 to *δ*_C_ 73.3 and the HMBC correlations of H_2_-6/C-4; H-5 (*δ*_H_ 1.36)/C-7, C-9, C-10, and C-6 (Figure 2). The absolute configuration of **6** was finally assigned by ECD spectrum (Figure 4), the calculated spectra showed the same trend as the experimental one, indicating the absolute configuration of **6** as 4*S*, 5*S,* and 10*R*.

Croargoid G (**7**) was obtained as a colorless gum. The HRESIMS ion peak at *m/z* 205.1199 ([M+Na]^+^, calcd. 205.1199) revealed its molecular formula as C_11_H_18_O_2_, with three DOUs. The ^13^C NMR and DEPT spectra (Table 2) of **7** revealed the presence of 11 carbons (three quaternary carbons, one methine, five methylene, and two methyl), which was different from the normal eudesmane type sesquiterpene carbon skeleton, and also different from the common monoterpene carbon skeleton. A detailed inspection of the 1D NMR and 2D NMR spectra signals revealed that ring A in **7** was the same as that of **5**, except that of ring B [16]. The existence of the five-membered ring B in compound **7** was confirmed by the HMBC correlations of H_2_-9 and H_2_-6/C-8 (*δ*_C_ 217.6); H-6/C-5, C-8, C-9, and C-10, and H_2_-9/C-5, C-6, and C-8 (Figure 2). Finally, the stereochemistry (4*R*, 5*R*, and 10*R*) of compound **7** was established by the calculated ECD spectrum.

### 2.2. NO Inhibitory and Cytotoxic Evaluations

All the isolates (**1**–**7**) were evaluated for their inhibitory effects on nitric oxide (NO) production stimulated by LPS in RAW 264.7 cells, with L-NMMA (N^G^-monomethyl-L-arginine, monoacetate salt) as a positive control. Compound **5** exhibited moderate NO inhibition and others were inactive at 50 *μ*M (Appendix A). In addition, their cytotoxic activity against five human cancer cell lines (HL-60, SMMC-7721, A-549, MCF-7, and SW-480) was also tested using the MTS method. However, all the compounds were inactive (IC_50_ > 40 *μ*M) (Appendix A).

## 3. Materials and Methods

### 3.1. General Experimental Procedures

HRESIMS spectra were obtained with a Shimadzu UPLC-IT-TOF mass spectrometer (Shimadzu Corp: Kyoto, Japan). The UV spectra were measured with a Shimadzu UV-2700 spectrophotometer (Shimadzu Corp: Kyoto, Japan). The IR spectra (KBr) were determined on a Nicolet iS10 spectrometer (Thermo Fisher Scientific: Waltham, MA, USA). Optical rotation was determined in MeOH on an Autopol VI polarimeter (Rudolph Research Analytical: Hackettstown, NJ, USA). The ECD spectra were recorded on a Chirascan circular dichroism spectrometer (Applied Photo Physics Ltd: Surrey, UK). NMR spectra were obtained on a Bruker Avance III 500 (Bruker Corp: Rheinstetten, Germany) with ^1^H NMR at 500 MHz and ^13^C NMR at 125 MHz using tetramethylsilane as internal standards. Semi-preparative HPLC was on a Waters 2695 system equipped with a YMC-Pack ODS-A column (250 × 10 mm, 5 *μ*m), using a flow rate of 3.0 mL/min at a column temperature of 28 °C, and detection was performed with a PDA detector. The silica gel GF254 (10~40 *μ*m) for TLC and silica gel (200–300 mesh) for column chromatography (CC) were produced from Qingdao Marine Chemical Factory: Qingdao, China. MCI gel (CHP20P, 75–150 *μ*m) was produced by Mitsubishi Chemical Corp: Kyoto, Japan.

### 3.2. Plant Material

The bark of *C. argyratus* was collected in Xishuangbanna Tropical Botanical Garden (XTBG), Chinese Academy of Sciences (CAS), Mengla County, Yunnan Province, China, in April 2021. They were identified by senior engineer Chun-Fen Xiao (one of the authors) of XTBG. A voucher specimen (No. HITBC-0032729) was deposited in the Herbarium of XTBG, CAS.

### 3.3. Extraction, Isolation, and Purification Process

Air-dried bark powder of *C. argyratus* (10 kg) was extracted three times with 95% EtOH (3 × 30 L, 3 days each time) to give a crude extract (1670 g), which was subjected to macroporous resin CC and eluted with a gradient system (MeOH/H2O, 30/60/90%), then collected the 90% fraction. The 90% fraction (960 g) was separated by a silica gel CC and eluted with gradient mixtures of petroleum ether/ethyl acetate (from 1:0 to 0:1) to obtain five fractions (A~E). This process was monitored using analytical TLC plates. The fraction C (188 g) was separated by MCI gel CC and eluted with MeOH/H2O (60/70/80/90/100%) to obtain fractions C-M1~C-M5. Thereafter, C-M1 was recrystallized to obtain **1** (4.0 g) and **2** (1.1 g), then the remaining C-M1 was purified by semi-preparative HPLC with 70% CH_3_CN/H_2_O as eluent to obtain **5** (300 mg, tR = 19 min) and **6** (5 mg, tR = 18 min). Fraction D was separated by MCI gel CC and eluted with MeOH/H_2_O (70/80/90%) to obtain three fractions (D-M1~D-M3). Fraction D-M1 (51.0 g) was subjected to silica gel CC and eluted with a gradient of petroleum ether/ethyl acetate to produce fractions D-M1-1~C-M1-11. Among them, D-M1-6 was separated by HPLC with 90% CH_3_CN/H_2_O to obtain **3** (2.5 g, tR = 9 min). Following the same procedure, **7** (5 mg, tR = 8 min) was obtained from fraction B-M2-1 by semi-preparative HPLC with 70% CH_3_CN/H_2_O.

### 3.4. Compound Characterization Data

Croargoid A (**1**): white amorphous powder; ${\left[a\right]}_{D}^{25}$ 129.8 (c 0.3, MeOH); UV (MeOH) *λ*_max_ (log*ε*) 223 (0.94) nm; IR (KBr) *ν*_max_ 3514, 2963, 2936, 1737, 1728, 1679 cm^−1^; ^1^H and ^13^C NMR date (Table 1 and Table 2); HRESIMS *m*/*z*: 273.1463 [M+Na]^+^, calcd. For C_15_H_22_O_3_Na^+^, 273.1461; CD (MeOH) *ν*_max_ (Δ*ε*) 225 (+29.12) nm; ^1^H and ^13^C NMR data, see Table 1 and Table 2.

Croargoid B (**2**): white amorphous powder; ${\left[a\right]}_{D}^{25}$ −206.1 (c 0.1, MeOH); UV (MeOH) *l*_max_ (log*e*) 196 (0.35), 222 (0.44) nm; IR (KBr) *ν*_max_ 3483, 2938, 2853, 1732, 1681 cm^−1^; ^1^H and ^13^C NMR date (Table 1 and Table 2); HRESIMS *m*/*z*: 295.1553 [M+HCOO]^-^, calcd. For C_16_H_23_O_5_^-^, 295.1551; CD (MeOH) *ν*_max_ (Δ*ε*) 224 (−28.08) nm; ^1^H and ^13^C NMR data, see Table 1 and Table 2.

Croargoid C (**3**): white amorphous powder; ${\left[a\right]}_{D}^{20}$ 141.7 (c 0.2, MeOH); UV (MeOH) *λ*_max_ (log*e*) 222 (0.44) nm; IR (KBr) *ν*_max_ 3566, 3392, 2985, 2961, 2925, 1738, 1692 cm^−1^; ^1^H and ^13^C NMR date (Table 1 and Table 2); HRESIMS *m*/*z*: 265.1446 [M-H]^-^, calcd. For C_15_H_21_O_4_^-^, 265.1445; CD (MeOH) *ν*_max_ (Δ*ε*) 241 (+35.71) nm; ^1^H and ^13^C NMR data, see Table 1 and Table 2.

Croargoid D (**4**): white amorphous powder; ${\left[a\right]}_{D}^{25}$ −48.4 (c 0.1, MeOH); UV (MeOH) *λ*_max_ (log*e*) 279 (0.94) nm; IR (KBr) *ν*_max_ 3530, 2985, 2917, 2866, 1769, 1666, 1679 cm^−1^; ^1^H and ^13^C NMR date (Table 1 and Table 2); HRESIMS *m*/*z*: 271.1303 [M+Na]^+^, calcd. For C_15_H_20_O_3_Na^+^, 271.1305; CD (MeOH) *ν*_max_ (Δ*ε*) 255 (+4.41), 288 (−9.11) nm; ^1^H and ^13^C NMR data, see Table 1 and Table 2.

Croargoid E (**5**): white amorphous powder; ${\left[a\right]}_{D}^{20}$ −90.0 (c 0.2, MeOH); UV (MeOH) *λ*_max_ (log*e*) 254 (0.51) nm; IR (KBr) *ν*_max_ 3434, 2979, 2967, 2937, 2922, 2862, 1659, 1588 cm^−1^; ^1^H and ^13^C NMR date (Table 1 and Table 2); HRESIMS *m*/*z*: 259.1668 [M+Na]^+^, calcd. For C_15_H_24_O_2_Na^+^, 259.1669; CD (MeOH) *ν*_max_ (Δ*ε*) 208 (−1.75), 221 (−0.28), 254 (−3.60), 284 (−0.41), 329 (−1.86) nm; ^1^H and ^13^C NMR data, see Table 1 and Table 2.

Croargoid F (**6**): white amorphous powder; ${\left[a\right]}_{D}^{20}$ 19.2 (c 0.1, MeOH); UV (MeOH) *λ*_max_ (log*e*) 232 (0.44) nm; IR (KBr) *ν*_max_ 3446, 2932, 2873, 2849, 1715, 1668 cm^−1^; ^1^H and ^13^C NMR date (Table 1 and Table 2) HRESIMS *m*/*z*: 259.1668 [M+Na]^+^, calcd. For C_15_H_24_O_2_Na^+^, 259.1669; CD (MeOH) *ν*_max_ (Δ*ε*) 243 (+13.16) nm; ^1^H and ^13^C NMR data, see Table 1 and Table 2.

Croargoid G (**7**): colorless gum; ${\left[a\right]}_{D}^{20}$ 126.4 (c 0.1, MeOH); UV (MeOH) *λ*_max_ (log*e*) 198 (0.35) nm; IR (KBr) *ν*_max_ 3486, 2996, 2954, 2933, 2864, 1733, 1613 cm^−1^; ^1^H and ^13^C NMR date (Table 1 and Table 2); HRESIMS *m*/*z*: 205.1199 [M+Na]^+^, calcd. For C_11_H_18_O_2_Na^+^, 205.1199; CD (MeOH) *ν*_max_ (Δ*ε*) 204 (−35.07), 297 (−40.16) nm; ^1^H and ^13^C NMR data, see Table 1 and Table 2.

### 3.5. Cell Culture and Nitric Oxide Inhibitory Assay

The RAW264.7 macrophages (obtained from Shanghai Cell Bank, Chinese Academy of Sciences: Shanghai, China) were maintained in Dulbecco’s modified Eagle’s medium (DMEM) (Shanghai Basal Media Technologies Corp., Ltd: Shanghai, China) at 37 °C in a humidity-constant incubator with 95% air and 5% CO_2_. RAW264.7 cells were seeded into 96-well plates at a concentration of 2 × 10^5^ cells/well and incubated for 24 h. After that, the cells were co-incubated with LPS (1 *μ*g/mL) (Sigma-Aldrich (Shanghai) Trading Co., Ltd: Shanghai, China). Then, the tested compounds (dissolved in DMSO) at 50 *μ*M concentrations were added into 96-well plates for incubating for 24 h, using L-NMMA (Sigma-Aldrich (Shanghai, China) Trading Co., Ltd: Shanghai, China) as a positive control. The cell viability was determined by MTS assay before the nitric oxide (NO) production assay, and the NO production was measured by the Griess Reagent System as previously reported [17].

### 3.6. Cytotoxicity Assay

The MTS method [18] was used for assessing the cytotoxicity of the compounds against five tumor cell lines (human myeloid leukemia HL-60, lung cancer A-549, hepatocellular carcinoma SMMC-7721, colon cancer SW480, and breast cancer MCF-7). All cells were cultured in DMEM medium containing 10% fetal bovine serum. Thereafter, 100 *μ*L cells (1 × 10^4^ cells/well) were seeded into 96-well plates and cultured for 12 h at 37 °C in a humidity-constant incubator with 95% air and 5% CO_2_ before adding the compounds. Then, the tested compounds (dissolved in DMSO, with 50 *μ*M concentrations) 100 *μ*L were added into 96-well plates for incubating for 48 h, each experiment was performed in triplicates with cisplatin as the positive control. After 48 h incubation, 20 *μ*L MTS solution and 100 *μ*L DMEM medium were added to each well and incubated for another 4 h. The OD value of each well was measured at 492 nm using a microplate reader (Multiskan FC, Thermo Fisher: Waltham, MA, USA).

## 4. Conclusions

In summary, seven new croargoid A–G (**1**–**7**) were isolated and characterized by solid data from *C. argyratus*. Compounds **1**–**4** were the first examples of eudesmane sesquiterpene lactones containing C_5_-OH group. Compound **7** was a highly degraded eudesmane sesquiterpene possessing a rare eleven-carbon skeleton. All isolates were evaluated for their cytotoxicities and NO inhibitions. Among those compounds, compound **5** exhibited moderate NO inhibition at 50 *μ*M.

## Figures and Tables

**Figure 1 molecules-27-06397-f001:**
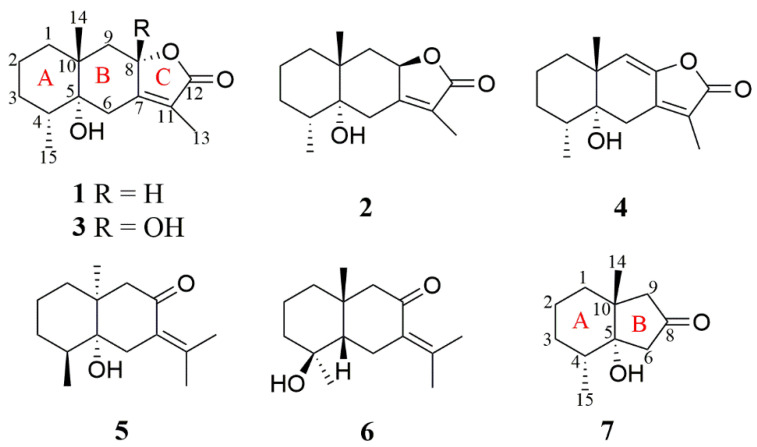
Structures of compounds **1**–**7**.

**Figure 2 molecules-27-06397-f002:**
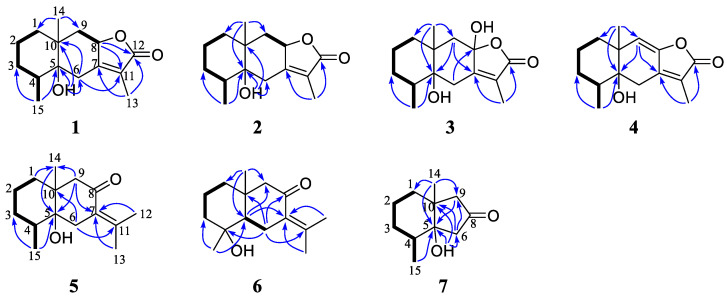
^1^H-^1^H COSY (━) and selected HMBC correlations (H→C) of compounds **1**–**7**.

**Figure 3 molecules-27-06397-f003:**
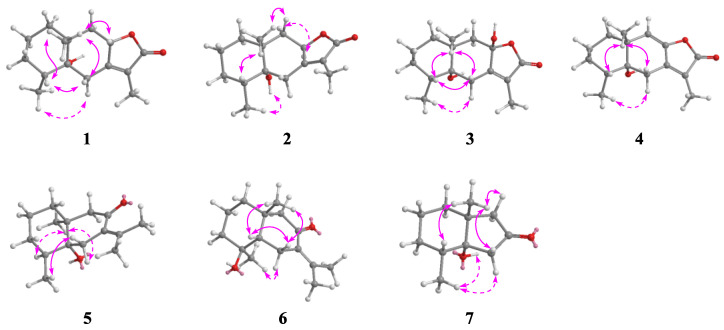
Key ROESY correlations (↔) of compounds **1**–**7**.

**Figure 4 molecules-27-06397-f004:**
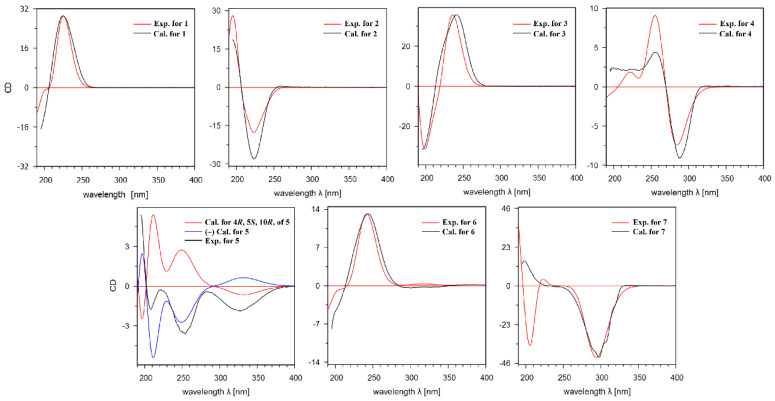
Experimental and calculated ECD spectra of compounds **1**–**7**.

**Table 1 molecules-27-06397-t001:** ^1^H (500 MHz) NMR data for compounds **1**–**7** in CDCl_3_ (*δ*_H_ in ppm, *J* in Hz).

No	1	2	3	4	5	6	7
1*α*	1.61, m ^a^	1.50, m ^a^	1.51, td (13.2,4.3)	1.35, m	1.16, d (12.2)	1.41, m ^a^	1.86, m
1*β*	1.15, m ^a^	1.35, m	1.09, td (13.2,4.3)	1.95, m ^a^	1.63, m	1.31, m ^a^	1.42, m
2*α*	1.48, m ^a^	1.49, m ^a^	1.44, m ^a^	1.65, m ^a^	1.54, m ^a^	1.42, m ^a^	1.58, m
2*β*	1.61, m ^a^	1.49, m ^a^	1.65, m	1.58, m ^a^	1.54, m ^a^	1.80, m	1.66, m
3*α*	1.46, m ^a^	1.24, m	1.30, m	1.24, m	1.48, m	1.69, m	1.38, m ^a^
3*β*	1.33, td (13.5,5.1)	1.49, m ^a^	1.44, m^a^	1.50, m ^a^	1.36, m	1.39, m ^a^	1.57, m ^a^
4	1.86, m ^a^	1.78, m ^a^	1.92, m	1.92, m ^a^	1.82, dd (8.5,6.7)		2.03, dd (13.4,6.7)
5						1.36, d (6.6)	
6*α*	2.80, d (14.5)	2.56, m	2.66, d (13.7)	2.87, d (17.4)	2.31, d (16.5)	3.00, d (16.7)	2.12, d (17.8)
6*β*	2.30, m	2.52, t (18.1)	2.38, d (13.7)	2.51, d (17.4)	2.69, d (16.5)	2.57, m	2.31, d (17.8)
8	4.93, m	5.39, m					
9*α*	1.88, m ^a^	2.27, dd (13.5,10.9)	1.89, d (13.6)	5.38, s	1.96, d (16.3)	3.28, d (15.5)	2.55, d (17.1)
9*β*	1.60, m ^a^	1.39, dd (13.5,5.9)	1.83, d (13.6)		2.67, d (16.3)	1.85, d (15.5)	1.90, d (17.1)
12					2.07, s	1.87, s	
13	1.81, s	1.79, s	1.76, s	1.90, s	1.79, s	1.99, s	
14	1.16, s	0.83, s	1.25, s	1.20, s	1.04, s	1.00, s	1.07, s
15	0.91, d (6.7)	0.93, d (6.8)	0.86, d (6.7)	0.96, d (6.7)	0.93, d (6.7)	1.30, s	0.86, d (6.7)
5-OH	1.24, s (br)	1.62, s					1.56, s

^a^ Overlapped signals.

**Table 2 molecules-27-06397-t002:** ^13^C (125 MHz) NMR data for compounds **1**–**7** in CDCl_3_ (*δ* in ppm).

No	1	2	3	4	5	6	7
1	34.3, CH_2_	36.9, CH_2_	34.1, CH_2_	32.9, CH_2_	35.1, CH_2_	40.0, CH_2_	30.2, CH_2_
2	20.1, CH_2_	21.1, CH_2_	19.9, CH_2_	20.7, CH_2_	21.1, CH_2_	18.0, CH_2_	20.4, CH_2_
3	30.1, CH_2_	30.2, CH_2_	29.9, CH_2_	29.8, CH_2_	30.5, CH_2_	40.5, CH_2_	29.3, CH_2_
4	34.8, CH	36.2, CH	34.0, CH	34.1, CH	35.0, CH	73.3, C	32.3, CH
5	76.7, C	76.4, C	77.7, C	75.8, C	74.1, C	48.2, CH	81.2, C
6	33.6, CH_2_	35.4, CH_2_	31.7, CH_2_	31.2, CH_2_	38.1, CH_2_	26.2, CH_2_	47.8, CH_2_
7	160.4, C	161.3, C	159.2, C	145.7, C	128.8, C	131.3, C	
8	78.3, CH	77.9, CH	104.1, C	148.2, C	203.1, C	205.7, C	217.6, C
9	42.8, CH_2_	43.2, CH_2_	46.8, CH_2_	116.1, CH	52.6, CH_2_	50.3, CH_2_	51.1, CH_2_
10	39.1, C	38.5, C	38.7, C	41.8, C	39.3, C	36.6, C	43.5, C
11	123.3, C	121.5, C	124.9, C	123.6, C	147.1, C	141.5, C	
12	174.9, C	175.3, C	173.3, C	171.3, C	23.8, CH_3_	22.3, CH_3_	
13	8.4, CH_3_	8.6, CH_3_	8.1, CH_3_	8.8, CH_3_	23.0, CH_3_	23.4, CH_3_	
14	20.7, CH_3_	23.9, CH_3_	21.0, CH_3_	24.0, CH_3_	22.3, CH_3_	30.8, CH_3_	23.0, CH_3_
15	15.2, CH_3_	14.9, CH_3_	15.1, CH_3_	15.2, CH_3_	15.2, CH_3_	31.4, CH_3_	16.2, CH_3_

## Data Availability

Not applicable.

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
