# Peer review of "Croargoids A–G, Eudesmane Sesquiterpenes from the Bark of Croton argyratus"

_molecules, 2022, doi:10.3390/molecules27196397_

Round 1

Reviewer 1 Report

The manuscript entitled "Croargoids A–G, Eudesmane Sesquiterpenes from the bark of Croton argyratus" presented by Xu and colleagues describes the efforts for the isolation and identification of new sesquiterpenes from C. argyrantus. The manuscript is well-written and concise. The biological relevance is still low, only two bio-assays are presented, with activities in the high micromolar range. However, the novelty surpasses the lack of biological activity of such compounds.

Reviewer 2 Report

Line 1-2:

Title needs to be revised as in this manuscript the author also includes cytotoxic and anti-inflammatory activity.

Recommendation: Cytotoxic and anti-inflammatory activities of Croargoids A–G, Eudesmane Sesquiterpenes from the bark of Croton argyratus

Line 33-34:C. argyratus is an important ethnic medicine, mainly distributed in Malaysia, Indonesia, Philippines, Vietnam, Thailand and other Southeast Asian countries” this statement can be mentioned in line 27-28. Seems like repetitive statement.

Line 46-226: Results explained in detail the isolation and identification of targeted compounds (1-7). No further suggestion and comments on this section. 

Line 175-176: Extraction using what apparatus? Rotary evaporator? Please state how long the dried bark been soaked in the 95% EtOH prior to the extraction process?

 Line 227-238: Be consistent in writing RAW264.7 / RAW 264.7 throughout the manuscript.

Line  228: What cell passage used?

Line 230: carbon dioxide (CO2), not CO2.

Line 233: LPS used (which brand and isolated from which source? E.coli? )

Line 233-234: Why the compound dissolved in DMSO and not directly using media? Any justification on this? And why the initial concentration used was 50 µM and not higher than this? Moreover, why there was no dose-dependent study shown in the results?   Any preliminary done prior to the assay? What is the concentration of L-NMMA used in the assay?

Line 237: Data Analysis should be in different section (3.7- Statistical Analysis). Which data are compared in this anti-inflammatory study? Isolated compounds vs positive control vs LPS? Please justify.

Line 239-250: How did you determined the IC50 of the cytotoxic activity as there is no results for dose-dependent study? Using Log Inhibition Curve? Justify.

Conclusion: OK

Tables:

Table S1. Anti-inflammatory study: Can the results shown in a histogram? Any explanation why the results for Compound 1-4 are having -% rate of NO inhibition? It is due to compound colouration that might interfere with the changes in Griess reagent, or the wavelength used in spectrophotometer? Please justify. How do you determine the strength of anti-inflammatory activity? Author mentioned that compound 5 showed weak inhibitory activity (41.62%) in line 147, however the L-NMMA showed 55%...almost similar. Please justify.

Reviewer 3 Report

Wu et al. isolated and comprehensively characterized 7 new croargoids, with one of the new compounds showing weak NO inhibitory activity. Some issues need to be fixed before the manuscript can be accepted for publication.

Line 26: Should be “are”

Line 51: Missing the word “groups”

Line 59: Please edit the sentence for clarity.

Line 117: Should be “determined”

Line 239: Please indicate the final DMSO concentration for the cytotoxicity assay.

Line 247: Should be “triplicates”

Line 252: Remove “ones”

Line 253: Should be “examples”
